# Effects of the Calix[4]arene Derivative Compound OTX008 on High Glucose-Stimulated ARPE-19 Cells: Focus on Galectin-1/TGF-β/EMT Pathway

**DOI:** 10.3390/molecules27154785

**Published:** 2022-07-26

**Authors:** Maria Consiglia Trotta, Francesco Petrillo, Carlo Gesualdo, Settimio Rossi, Alberto Della Corte, Judit Váradi, Ferenc Fenyvesi, Michele D’Amico, Anca Hermenean

**Affiliations:** 1Department of Experimental Medicine, University of Campania “Luigi Vanvitelli”, 80138 Naples, Italy; mariaconsiglia.trotta2@unicampania.it (M.C.T.); michele.damico@unicampania.it (M.D.); 2PhD Course in Translational Medicine, Department of Experimental Medicine, University of Campania “Luigi Vanvitelli”, 80138 Naples, Italy; francescopetrillo09@gmail.com; 3Multidisciplinary Department of Medical, Surgical and Dental Sciences, Eye Clinic, University of Campania “Luigi Vanvitelli”, 80138 Naples, Italy; carlo.gesualdo@unicampania.it (C.G.); settimio.rossi@unicampania.it (S.R.); albertodellacorte@live.it (A.D.C.); 4Department of Pharmaceutical Technology, Faculty of Pharmacy, University of Debrecen, Nagyerdei St. 98, H-4032 Debrecen, Hungary; varadi.judit@pharm.unideb.hu (J.V.); fenyvesi.ferenc@pharm.unideb.hu (F.F.); 5Faculty of Medicine, Vasile Goldis Western University of Arad, 310414 Arad, Romania

**Keywords:** diabetic retinopathy, retinal pigment epithelial cells, galectin-1, OTX008, fibrosis, epithelial-mesenchymal transition

## Abstract

Diabetic retinopathy (DR) is a neurovascular disease characterized by the reduction of retina integrity and functionality, as a consequence of retinal pigment epithelial cell fibrosis. Although galectin-1 (a glycan-binding protein) has been associated with dysregulated retinal angiogenesis, no evidence has been reported about galectin-1 roles in DR-induced fibrosis. ARPE-19 cells were cultured in normal (5 mM) or high glucose (35 mM) for 3 days, then exposed to the selective galectin-1 inhibitor OTX008 (2.5–5–10 μM) for 6 days. The determination of cell viability and ROS content along with the analysis of specific proteins (by immunocytochemistry, Western blotting, and ELISA) or mRNAs (by real time-PCR) were performed. OTX008 5 μM and 10 μM improved cell viability and markedly reduced galectin-1 protein expression in cells exposed to high glucose. This was paralleled by a down-regulation of the TGF-β/, NF-kB p65 levels, and ROS content. Moreover, epithelial–mesenchymal transition markers were reduced by OTX008 5 μM and 10 μM. The inhibition of galectin-1 by OTX008 in DR may preserve retinal pigment epithelial cell integrity and functionality by reducing their pro-fibrotic phenotype and epithelial–mesenchymal transition phenomenon induced by diabetes.

## 1. Introduction

Recent results suggest galectin-1 (Gal-1) as a new possible therapeutic target for fibrosis in diabetes [1]. Gal-1 belongs to a group of proteins binding β-galactoside sugars by N-linked or O-linked glycosylation and has been associated to cell adhesion/proliferation, immune responses, apoptosis, and inflammation [2]. Gal-1 was found upregulated in the intraocular microenvironment of diabetic retinopathic patients along with the progression of clinical stages of retinopathy but not in the intraocular microenvironment of non-diabetic retinopathic patients [3], indicating a role for Gal-1 in diabetic retinopathy (DR) and diabetes or related hyperglycemia, a primum movens in its production.

Similarly, there is increasing evidence that transforming growth factor beta (TGF-β) signaling plays a critical role in DR. In fact, TGF-β1 expression in DR patients, DR rats, and high glucose-incubated human retinal endothelial cells (HRECs) and human adult retinal pigment epithelial cells (ARPE-19) is increased [4,5,6,7], possibly representing a target of anti-fibrotic therapy during DR.

Worthy of note is that fibrosis in DR is an ultimate process of the proliferative form that determines retinal detachment. It involves Muller cells, astrocytes, microglial cells [8], and retinal pigment epithelial cells (RPE) [9]. Among these, RPE undergo transformation to fibroblast-like cells, proliferate, and produce extracellular matrix deposition and expansion, leading the formation of fibrotic tissue through an epithelial–mesenchymal transition (EMT) [10,11] that reduces the retina’s integrity and flexibility. Fibrosis may compromise vision and ultimately lead to blindness. Therefore, identifying the pattern of mediators involved in DR-induced fibrosis can be helpful.

In this context, while the data described above delineate a role of Gal-1 in fibrosis and DR, there are no data supporting Gal-1 in the pro-fibrotic phenotype assumed by RPE under continuous high-glucose stimulus. Similarly, there are no data on the effects of Gal-1 inhibition in relation to the high-glucose-stimulated TGF-β/EMT pathway in ARPE-19.

On another note, inhibition of Gal-1 by the calix[4]arene derivative compound OTX008, showing a half-maximal inhibitory concentration (IC50) from 1 to 190 µM against Gal-1, has been experimented for tumor targeting [12,13,14]. Calixarenes and related macrocycles have been also shown to exert an antimicrobial activity by inhibiting the biofilm formation [15] and have been extensively tested for their application in drug/gene delivery, in vitro, or in vivo diagnosis and therapy [16]. To this regard, OTX008 has been proposed as a possible therapeutic tool to control fibrosis in proliferative diabetic retinopathy [17].

Here, therefore, we resolved to study the cellular expression of Gal-1 in physiological or high-glucose conditions on a cell line highly involved in the DR-induced fibrotic process, the human ARPE-19 cell line, that maintains the feature of the RPE in vivo. Then, on these cells, we assessed whether Gal-1 enhances TGF-β signaling in a positive feedback loop and assessed the effects of blocking the TGF-β/EMT pathway with the OTX008 Gal-1 inhibitor.

## 2. Results

### 2.1. Cell Viability

The exposition of ARPE-19 cells to high glucose (35 mM, HG) significantly altered their morphology: they were elongated under normal glucose (5 mM, NG), appearing densely packed in colonies, and meanwhile became irregular and shrunken in HG, characterized by dissociation of cell colonies. HG also reduced the percentage of cell viability (46 ± 7%, *p* < 0.01 vs. NG) compared to cells exposed to NG alone (NG; 90 ± 4%) or with mannitol 30 mM (NG + M; 87 ± 7%). HG cells exhibited a significant increment in cell viability when stimulated with OTX008 5 µM (HG + OTX 5; 59 ± 6%, *p* < 0.05 vs. HG) and 10 µM (HG + OTX 10; 60 ± 2%, *p* < 0.05 vs. HG) (Figure 1).

### 2.2. Gal-1 Expression

A significant increase of adaptor-related protein complex 4 subunit sigma 1 (AP4S1), the transcription factor able to promote Gal-1 expression, was evident in HG cells both as mRNA (2.6 ± 0.3 2^^−ΔΔCt^, *p* < 0.01 vs. NG) and protein levels (1086 ± 130 pg/mL; *p* < 0.01 vs. NG) compared to NG (1.2 ± 0.1 2^^−ΔΔCt^, 494 ± 49 pg/mL) or NG + M (1.0 ± 0.2 2^^−ΔΔCt^, 493 ± 73 pg/mL) treatments. OTX008 did not significantly modify AP4S1 cell content (2.5 µM: 2.4 ± 0.4 2^^−ΔΔCt^, 1065 ± 107 pg/mL; 5 µM: 2.5 ± 0.4 2^^−ΔΔCt^, 945 ± 77 pg/mL; 10 µM: 2.8 ± 0.4 2^^−ΔΔCt^, 952 ± 77 pg/mL) (Figure 2).

**Figure 2 molecules-27-04785-f002:**
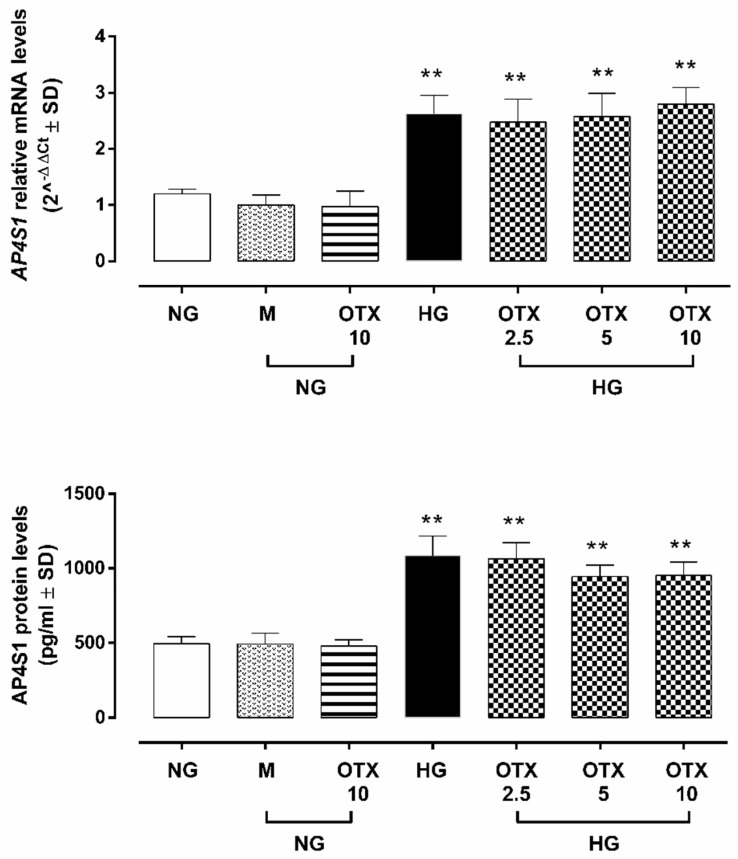
AP4S1 expression in ARPE-19 cells. *AP4S1* relative mRNA levels, reported as 2^^−ΔΔCt^ ± SD and quantized by using *GAPDH* as control along with AP4S1 protein levels, reported as pg/mL  ±  SD. N = 9 observations per group (three independent experiments, each performed in triplicate). ** *p* < 0.01 vs. NG; NG, normal glucose (5 mM) + vehicle; M, mannitol (30 mM); HG, high glucose (35 mM) + vehicle; OTX 2.5, OTX 2.5 µM; OTX 5, OTX 5 µM; OTX 10, OTX 10 µM.The up-regulation of AP4S1 was paralleled by a significant increase of Gal-1 mRNA (*LGALS1*) in HG cells (3.2 ± 0.3 2^^−ΔΔCt^, *p* < 0.01 vs. NG), which was not reverted by OTX treatments (2.5 µM: 3.0 ± 0.4 2^^−ΔΔCt^; 5 µM: 3.1 ± 0.2 2^^−ΔΔCt^; 10 µM: 3.4 ± 0.4 2^^−ΔΔCt^) (Figure 3A). Accordingly, Gal-1-positive cells in ARPE-19 exposed to HG (66 ± 10%; *p* < 0.01 vs. NG), which was absent in ARPE-19 cells exposed to NG alone (13 ± 4%) or with mannitol (19 ± 7%). HG cells stimulated with OTX008 5 µM and 10 µM exhibited a significant reduction in Gal-1-positive stained cells (OTX 5: 45 ± 4% and OTX 10: 41 ± 7%; both *p* < 0.01 vs. HG) (Figure 3B). Accordingly, Gal-1 protein levels were significantly increased in HG cells (Gal-1/β-actin = 1.01 ± 0.1, *p* < 0.01 vs. NG), while they were reduced in HG cells exposed to OTX008 5 µM (Gal-1/β-actin = 0.39 ± 0.04, *p* < 0.01 vs. HG) and 10 µM (Gal-1/β-actin = 0.42 ± 0.04, *p* < 0.01 vs. HG) (Figure 3C). OTX alone did not modify the Gal-1 expression in NG-treated ARPE-19 even at the highest concentration of 10 µM.

### 2.3. TGF-β Signaling

ARPE-19 cells exhibited a significant up-regulation of *TGF-β1* (4.8 ± 0.9 2^^−ΔΔCt^; *p* < 0.01 vs. NG), transforming growth factor β receptor 1 (*TGF-βR1*: 2.3 ± 0.5 2^^−ΔΔCt^; *p* < 0.05 vs. NG), and transforming growth factor β receptor 2 (*TGF-βR2*: 2.8 ± 0.3 2^^−ΔΔCt^; *p* < 0.01 vs. NG) mRNA levels (Figure 4).

Conversely, ARPE-19 cells exposed to 5 µM and 10 µM OTX008 treatments exhibited a significant reduction of *TGF-β1* (respectively, 3.1 ± 0.2 and 2.9 ± 0.3 2^^−ΔΔCt^; both *p* < 0.01 vs. HG), *TGF-βR1* (respectively, 1.6 ± 0.3 and 1.6 ± 0.2 2^^−ΔΔCt^; both *p* < 0.05 vs. HG), and *TGF-βR2* (respectively, 1.8 ± 0.2 and 1.7 ± 0.4 2^^−ΔΔCt^; both *p* < 0.01 vs. HG) mRNA levels (Figure 4).

### 2.4. NF-kB p65, IkB-β, and ROS Pathway

Higher levels of TGF-β were paralleled by a higher percentage of nuclear factor kappa light-chain enhancer of activated B cells p65 (NF-kB p65)-positive cells (69 ± 10%; *p* < 0.01 vs. NG) and NF-kB p65 protein expression (NF-kB p65/β-actin = 2.2 ± 0.3, *p* < 0.01 vs. NG) in HG cells. A significant reduction of NF-kB p65-positive cells and protein levels was observed in HG cells treated with OTX008 5 µM (NF-kB p65-positive cells: 33 ± 6%; NF-kB p65/β-actin: 1.4 ± 0.2, both *p* < 0.01 vs. HG) and 10 µM (NF-kB p65-positive cells: 25 ± 3%; NF-kB p65/β-actin: 1.2 ± 0.2, both *p* < 0.01 vs. HG) (Figure 5A,B). Conversely, the protein levels of inhibitor of NF-kB subunit beta (IkB-β) were markedly reduced in ARPE-19 cells by HG exposure (IkB-β/β-actin: 0.5 ± 0.08, *p* < 0.01 vs. NG) but significantly higher in HG cells treated with OTX008 at the doses of 5 µM (IkB-β/β-actin: 0.9 ± 0.1, *p* < 0.01 vs. HG) and 10 µM (IkB-β/β-actin: 1.0 ± 0.1, *p* < 0.01 vs. HG) (Figure 5B).

Accordingly, reactive oxygen species (ROS) levels were significantly increased in HG cells (67 ± 4%; *p* < 0.01 vs. NG), with ROS content significantly reduced by OTX008 treatments at doses of 5 µM (48 ± 8%, *p* < 0.01 vs. HG) and 10 µM (46 ± 4%, *p* < 0.01 vs. HG) (Figure 6).

### 2.5. EMT

ARPE-19 cells exposed to HG exhibited increased levels of smooth muscle actin alpha 2 mRNA and protein levels (*ACTA2*: 3.0 ± 0.1 2^^−ΔΔCt^; 114 ± 17 ng/mL; *p* < 0.01 vs. NG) as well as fibronectin I (*Fn1*: 2.8 ± 0.3 2^^−ΔΔCt^; 97 ± 11 ng/mL; *p* < 0.01 vs. NG), two markers of the EMT process in RPE cells. Both were reduced in HG cells by OTX008 at the doses of 5 µM (ACTA2: 2.1 ± 0.3 2^^−ΔΔCt^, *p* < 0.01 vs. HG; 79 ± 13 ng/mL; *p* < 0.05 vs. HG; Fn1: 1.9 ± 0.1 2^^−ΔΔCt^; 61 ± 9 ng/mL; *p* < 0.01 vs. HG) and 10 µM (ACTA2: 2.1 ± 0.2 2^^−ΔΔCt^, *p* < 0.01 vs. HG; 74 ± 15 ng/mL; *p* < 0.05 vs. HG; Fn1: 1.8 ± 0.1 2^^−ΔΔCt^; 62 ± 8 ng/mL; *p* < 0.01 vs. HG) (Figure 7).

## 3. Discussion

Diabetic retinopathy is generally considered a debilitating microvascular complication, and the effects on the RPE have received much less attention. RPE forms the outer blood–retinal barrier between the choroid and neurosensory retina to protect the retina from damaging stimuli [9], and in diabetic retinopathy’s later stages, it begins to settle on the retina, transform into fibroblast-like cells, and disassemble in a proliferative fibrotic phenotype. Some controversies still exist on the pattern of mediators and pathways involved in this, and in-vivo-like RPE culture models may identify new targets and tools for preventing how diabetes affects RPE-specific functions.

Here, it is reported for the first time that OTX008 (or PTX-008), a calixarene-based compound and galectin-1 (Gal-1) inhibitor, is able to block the deleterious effects exerted by high concentrations of glucose on the cell viability of ARPE-19 cells. Interestingly, the compound reduced the burst of free radicals released into the medium by hyper glucose. Effects paralleled by the reduction of an important marker in these cells, the Gal-1, were in line with the reduction of Gal-1 expression previously reported in hypoxic human retinal Müller glial cells (HRMECs) exposed to OTX008 [17]. Interestingly, the pro-fibrotic cytokine TGF-β, hyper-expressed by high glucose, was also reduced by the treatment of ARPE-19 cells with OTX008.

Calix[n]arenes are synthetic skeletons that support functionalization of several molecules and derivatives applied in the biomedical field as drugs [18]. They function as nanocarriers for drugs delivery and optimize their activities [19,20,21]. Particularly, OTX008, N-[2-dimethylamino)ethyl] acetamidyl calix[4]arene, previously showed to bind Gal-1 and lead to its oxidation and proteosomal degradation through a mechanism that is not yet elucidated, displayed anti-angiogenic and antineoplastic activities by Gal-1 and Erk1/2 and protein kinase B (AKT)-dependent survival pathways inhibition, and induced G2/M cell cycle arrest through cyclin-dependent kinase 1 (CDK1) [13,22].

Worthy of note is that Gal-1 is considered a new fibrotic promoter protein in type 1 and type 2 diabetes [1], whose production is activated by the AP4 transcription factor through AKT pathway and is translated in excess deposition of the extracellular matrix (ECM). Fibroblasts are the main cells responsible for regulating ECM homeostasis along with other mesenchymal cell types, such as mesangial cells in the kidney and stellate cells in the liver [23]. The activation, proliferation, and persistence of those cells are mediated by cytokines, chemokines, and growth factors, such as TGF-β, platelet-derived growth factor (PDGF) and matrix metalloproteinases (MMPs) [24]. Among them, TGF-β signaling is considered the key profibrogenic pathway, which promotes the secretion of ECM components and deposition and induces terminal differentiation of fibroblasts into myofibroblasts and tissue contraction [25].

Pioneer studies done in a clinical setting by Connor et al. [26] and later confirmed by ref. [27] evidenced the strong and pushing role of TGF-β in ocular fibrosis, with levels of the cytokines as high as three times that in the non-fibrotic eye. Recently, Gal-1-knockout mice and RPE cells have linked the TGF-β to Gal-1 as two pathways talking each other to control a series of fibrosis transducers such as, for example, type 1 collagen and Fn1 [28]. OTX008, which is a well-established inhibitor of Gal-1, was able to counteract the putative actions derived from these agents, as seen in the various settings tested on ARPE-19 cells, through its proteasomal degradation of Gal-1 protein and not through its ubiquitination [1].

Signaling downstream of TGF-β includes canonical (through SMAD) or non-canonical pathways through Erk, MAPK, NF-kB, and AKT, which crosstalk with SMAD [23]. OTX008 was able to counteract the TGF-β and the putative actions derived from it, as seen in the various settings tested on ARPE-19 cells.

The role played by inflammation in the genesis of fibrosis is widely known, with a major role driven by interleukins (IL)-1β, IL-6, IL-8, tumor necrosis factor-alpha (TNFα), and NF-kB [29].

In line with this, OTX008 restored the content of IkB-β, the NF-kB protein blocker, while decreasing the levels of the transcription factor. These evidences are in line with the attenuation of NF-kB in hypoxic HRMECs exposed to OTX008 [17].

Another aspect of the OTX008 that, however, has to be considered for future perspective studies is that by blocking the Gal-1/TGF-β pathway, the compound may candidate itself as a drug conditioning the EMT process of ARPE-19 cells under high glucose. Indeed, here, it is shown that the canonical markers of EMT such as, for instance, ACTA2 and FnI [10,30] are reduced by the treatment of the cells with OTX008. Worthy of note is that the EMT of polarized RPE cells is a process activated under pathological circumstances, such as inflammation, wound healing, and carcinogenesis, enabling epithelial cells to obtain an enhanced migration ability and increase their production of extracellular matrix components, leading to RPE dysfunction [31]. Loss of RPE functional integrity is the base for development of numerous retinal diseases, including inherited cone-rod degenerations, inherited macular degeneration, age-related macular degeneration, and proliferative vitreoretinopathy [28]. OTX008, by an activity triggered by Gal-1 inhibition, may orchestrate and direct various actors on the complex scenario of high-glucose-induced damage of ARPE-19 cells towards a stabilization of their physiological functions (e.g., prevention of vision loss and photoreceptor integrity preservation) instead of pathological ones.

The authors do not exclude that OTX may have additional effects to the inhibition of Gal-1, but it is not possible to know this completely due to the few studies done on the compound and given its recent introduction into experimentation. Similarly, it is not clear whether Gal-1 is expressed by other cells of the retina involved in fibrosis, and thus, in vivo studies or in vitro studies on different cell lines may corroborate the concept expressed in the present study.

In conclusion, OTX008, by preventing the pro-fibrotic processes activated by Gal-1 in a cell line strictly involved in the pathogenic mechanisms of diabetes- or hyperglycemia-related eye diseases, candidates itself as good therapeutic tool to treat them.

## 4. Materials and Methods

### 4.1. Cell Culture

The human ARPE-19 cell line, obtained from the American Type Culture Collection (ATCC), was cultured at 37 °C and 5% CO_2_ in Dulbecco’s modified Eagle’s medium/nutrient mixture F12 (DMEM/F12, Aurogene AU-L0093), with a glucose concentration of 5 mM (Life Technologies A24940-01). The medium was supplemented with 10% heat-inactivated fetal bovine serum (FBS) (AU-S181H Aurogene, Italy), 1% penicillin/streptomycin solution (P/S) (Aurogene Au-L0022), 1% L-Glutamine (L-Glu), Hepes 5 mM (Thermo Fisher 15630080), and 7.5% NaHCO3 (Thermo Fischer 25080094). Two days after seeding, cells were split and plated at different cell densities for each assay. Cells were then exposed for the first 3 days at 37 °C and 5% CO_2_ to 5 mM d-glucose (normal glucose); 35 mM d-glucose (high glucose) [32]; and 5 mM d-glucose and 30 mM mannitol (NG + M) (Sigma 69-65-8) as negative control for high-glucose concentration [32]. Then, after 72 hours of NG or HG stimulation, OTX008 (C_52_H_72_N_8_O_8_, chemical structure presented in Appendix A—S6949, Selleckchem, Houston, TX 77014 USA) dissolved in 0.1% w/v dimethyl sulfoxide (DMSO) as vehicle or DMSO 0.1% was daily added to fresh cell medium (NG or HG) until day 9 [33]. Particularly, ARPE-19 cells were exposed to OTX008 10 µM, which was previously effective in human retinal cells [17] and at lower OTX concentrations (2.5 and 5 µM), as follows:I.5 mM d-glucose + vehicle (NG);II.5 mM d-glucose + OTX008 10 µM (NG + OTX10);III.35 mM d-glucose + vehicle (HG);IV.35 mM d-glucose + OTX008 2.5 µM (OTX 2.5);V.35 mM d-glucose + OTX008 5 µM (OTX 5);VI.35 mM d-glucose + OTX008 10 µM (OTX 10).

For the daily observation of ARPE-19 morphology with optic microscope (Leica DMi1, Mannheim, Germany), cells were plated in 6-well plates at a density of 2 × 10^5^ cells/well as well as for protein and RNA isolation [34]. After the stimulation period, cells and supernatants were collected and preserved for down-stream analysis. For each assay, three independent experiments were performed, each done in triplicate.

### 4.2. Cell Viability Assay

Cell viability was measured by 3-(4,5-dimethylthiazol2-yl)-2,5-diphenyltetrazolium bromide (MTT) assay. Particularly, ARPE-19 (6 × 10^3^ cell/well) were seeded in 96-well plates [32] and exposed to normal or high glucose, with or without OTX, as previously described. At the end of the stimulation period, MTT solution (1:10 in culture medium, 300 µL/well) was added to each well, incubated for 4 h at 37 °C, and then removed. Each well was then washed for 20 min with isopropanol-HCl 0.2 N. Optical density (OD) values were measured at 570 nm using a 96-well plate reader (iMark, Bio-Rad Laboratories, CA, USA) [35].

### 4.3. ROS Assessment

ROS levels were detected by the conversion of the fluorescent probe 2′,7′-dichlorodihydrofluorescein diacetate (DCFH-DA) to highly fluorescent dichlorofluorescein (DFC) diacetate within cells by ROS. ARPE-19 were seeded in 96-well plates (5 × 10^3^ cells/well) [36] and exposed to normal or high glucose, with or without OTX. At the end of the stimulation period, cells were loaded with 20 µM DCFH-DA in medium with 5% FBS at 37 °C for 30 min and then were trypsinized. Total intracellular ROS production was measured with a fluorometric plate reader at an excitation of 485 nm and an emission of 530 nm. Both cell types were exposed to medium 5% FBS without DCFH-DA as negative control (CTR−) or incubated with H_2_O_2_ (100 µM) 30 min before trypsinization as a positive control (CTR+) [32].

### 4.4. Immunocytochemistry

For immunocytochemical analysis, ARPE-19 (3 × 10^3^ cells/well) were plated on slides in 24-well plates [37] and exposed to normal or high glucose, with or without OTX. Cells were fixated with 4% paraformaldehyde and washed with phosphate-buffered saline (PBS) (AU-L0615 Aurogene, Italy). Then, in order to inhibit non-specific antibody binding, cells were incubated for 60 min in blocking solution with 3% bovine serum albumin (BSA) (Sigma A7906) and 0.3% Triton X-100 (Sigma 93443) in PBS [38]. Primary antibodies were diluted in PBS-blocking buffer (1% BSA, 0.1% Triton X-100), and slides were incubated overnight at 4 °C in primary antibodies to human Gal-1 (Thermo Fisher PA5-95213; 0.7 µg/mL, to rabbit) and to human NF-kB p65 (Cell Signaling 6956S; dilution 1:500, to mouse).

Fluorescent-labeled anti-rabbit (Invitrogen 11,008; dilution 1:1000) and anti-mouse (Life Technologies A21202; dilution 1:1000) secondary antibodies were used to locate the specific antigens in each slide. Cells were counterstained and mounted with VECTASHIELD Antifade Mounting Medium with 4′,6-diamidino-2-phenylindole (DAPI) (Novus Biologicals H-1200-NB). Fluorescently labeled slides were viewed with a fluorescence microscope (Leica, Wetzlar, Germany) and with a fluorescence confocal microscope (LSM 710, Zeiss, Oberkochen, Germany). Immunofluorescence images were analyzed with Leica FW4000 software (Leica, Wetzlar, Germany) and with Zen Zeiss software (Zeiss, Oberkochen, Germany). The percentage of positive cells in each microscope field was calculated by the number of positive cells of 350 cells in four different microscope fields for each treatment by considering only DAPI counterstained cells as positive profiles. Data were reported as mean percentage of positive cells/total cells counted ± standard deviation (SD).

### 4.5. Enzyme-Linked Immunosorbent Assay (ELISA)

Competitive ELISA tests were used to quantify the cellular levels of human AP4S1 (My Biosource, MBS7206462), ACTA2 (or α-SMA) (antibodies-online, ABIN6953406), and Fn1 (Cusabio, CSB-E11850h).

### 4.6. Western Blotting

After trypsinization, ARPE-19 cells were resuspended in RIPA buffer (Sigma, R0278) containing protease and phosphatase inhibitors to isolate protein content. After centrifuging samples at 12,000 rpm for 10 min at 4 °C, protein levels in the supernatants were determined by using a Bio-Rad Protein Assay (Bio-Rad Laboratories, 500-0006). Western blotting analysis was performed as previously described [36]. Briefly, proteins were separated on a 10% sodium dodecyl sulfate polyacrylamide gel electrophoresis (SDS–PAGE) and then electrotransferred to polyvinylidene difluoride (PVDF) membranes (Merck Millipore, IPFL10100). Membranes were blocked for 1 h with a 5% non-fat dry milk/Tris-buffered saline (TBS) solution (Euroclone, EMR180500; Cell Signaling 12498) before the incubation at 4 °C over-night with the following human primary antibodies: anti-Gal-1 (Thermo Fisher PA5-95213; 0.3 µg/mL in 3% blocking solution, to rabbit), anti-NF-kB p65 (Cell Signaling 6956S; dilution 1:1000 in 3% blocking solution, to mouse), and IkB-β (Santa Cruz sc-945; dilution 1:1000 in 3% blocking solution, to rabbit). Horseradish peroxidase-conjugated secondary anti-rabbit (Santa Cruz, sc-2004; dilution 1:5000 in 3% blocking solution) or anti-mouse (Santa Cruz, sc-2005; dilution 1:5000 in 3% blocking solution) antibodies were used to incubate the blots at room temperature for 1 h. Immunoreactive bands were visualized by using an enhanced chemiluminescence system (Thermo Fisher, 35055), then quantified with VisionWorks Life Science Image Acquisition and Analysis software (UVP, Upland, CA, USA) and expressed as densitometric units (DU). Gal-1, NF-kB, and IkB-β protein levels were normalized by using β-actin protein levels (Santa Cruz sc-47778, dilution 1:1000 in 3% blocking solution, to mouse) (uncropped images presented in Appendix A).

### 4.7. Real-Time Quantitative Reverse Transcription PCR (qRT-PCR)

Total RNA was isolated from ARPE-19 lysates following the miRNeasy Mini kit (Qiagen, 21,7004). RNA concentration and purity was determined by using the NanoDrop 2000c Spectrophotometer (Thermo Fisher Scientific, Waltham, MA, USA). Genomic DNA (gDNA) contaminations were eliminated from RNA samples before the reverse transcription (RT) step, carried out on the Gene AMP PCR System 9700 (Applied Biosystems, MA, USA) by using the QuantiTect Reverse Transcription kit (20,5311, Qiagen), according to the protocol “Reverse Transcription with Elimination of Genomic DNA for Quantitative, Real-Time PCR”. The final step for real-time PCR (qPCR) analysis was carried out in triplicate on the CFX96 Real-time System C1000 Touch Thermal Cycler (Biorad). This was performed according to the protocol “Two-Step RT-PCR (Standard Protocol)” by using the QuantiTect SYBR Green PCR Kit (20,4143, Qiagen) and specific QuantiTect Primer Assays (24,9900, Qiagen) for each gene tested (*TGF-β1*: QT00000728; *TGF-βR*1: QT00083412; *TGF-βR2*: QT00014350; *AP4S1: QT00197001; LGALS1: QT00064113; ACTA2*: QT00088102; and *Fn1*: QT00038024). Relative quantization of gene expression was performed by using the 2^^−ΔΔCt^ method [39] by using glyceraldehyde 3-phosphate dehydrogenase (*GAPDH*) (QT00079247) as housekeeping control gene.

### 4.8. Statistical Analysis

The results are reported as mean ± SD of three independent experiments, with each performed in triplicate. Statistical significance was determined using one-way analysis of variance (ANOVA) followed by Tukey’s comparison test by using GraphPad Prism 6.0. A *p*-value less than 0.05 was considered significant to reject the null hypothesis.

## Figures and Tables

**Figure 1 molecules-27-04785-f001:**
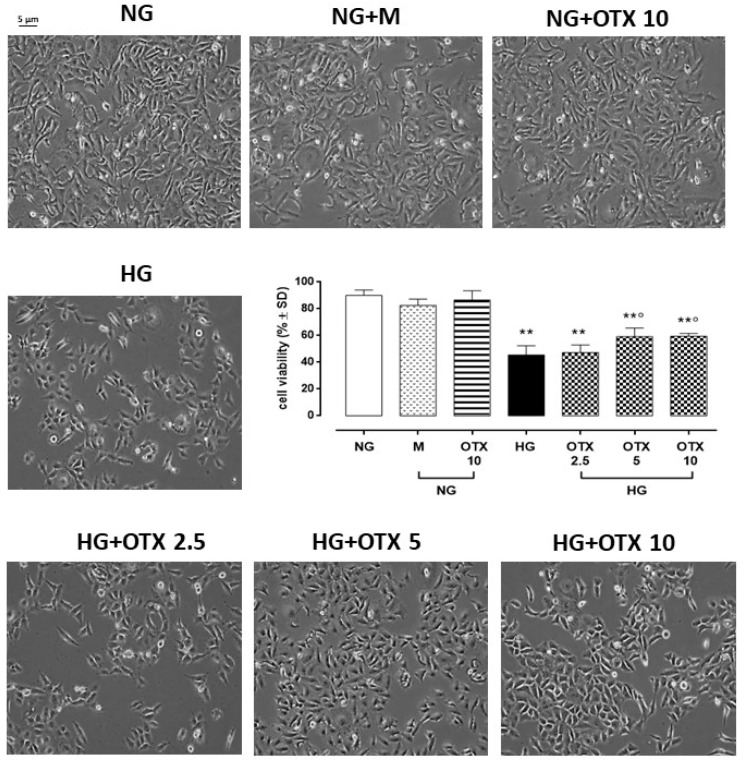
Representative optic microscope observations and cell viability of ARPE-19. For MTT assay, cells were plated at a density of 6 × 10^3^ cells/well in 96-well plates and cultured for 3 days in NG or HG. OTX was added starting from day 4 until day 9. Results are reported as a percentage of cell viability ± SD of 9 observations per group (three independent experiments, each performed in triplicate). ** *p* < 0.01 vs. NG; ° *p* < 0.05 vs. HG; NG, normal glucose (5 mM) + vehicle; M, mannitol (30 mM); HG, high glucose (35 mM) + vehicle; OTX 2.5, OTX 2.5 µM; OTX 5, OTX 5 µM; OTX 10, OTX 10 µM. Magnification = 10×; scale bar = 5 µM.

**Figure 3 molecules-27-04785-f003:**
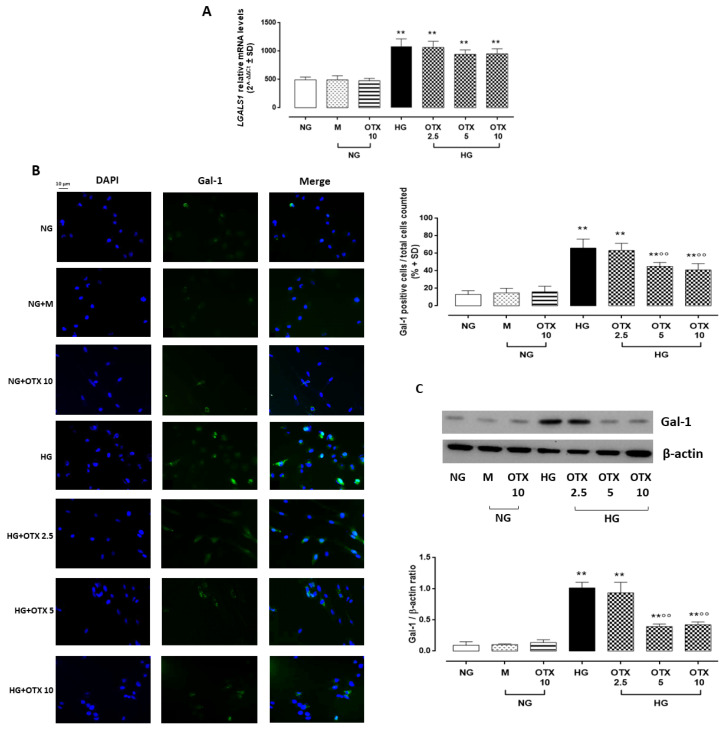
Gal-1 levels in ARPE-19 cells. (**A**) *LGALS1* relative mRNA levels, reported as 2^^−ΔΔCt^ ±  SD and quantized by using *GAPDH* as control. (**B**) Representative immunofluorescence images of Gal-1-positive staining, reported as a percentage of Gal-1-positive cells/total cells counted ± SD. Magnification = 10×; scale bar = 10 µM. (**C**) Representative Western blotting dots of Gal-1 and β-actin. Protein levels, quantified as densitometric units ± SD, are reported as Gal-1/β-actin ratio. N = 9 observations per group (three independent experiments, each performed in triplicate) ** *p* < 0.01 vs. NG; °° *p* < 0.01 vs. HG; NG, normal glucose (5 mM) + vehicle; M, mannitol (30 mM); HG, high glucose (35 mM) + vehicle; OTX 2.5, OTX 2.5 µM; OTX 5, OTX 5 µM; OTX 10, OTX 10 µM.

**Figure 4 molecules-27-04785-f004:**
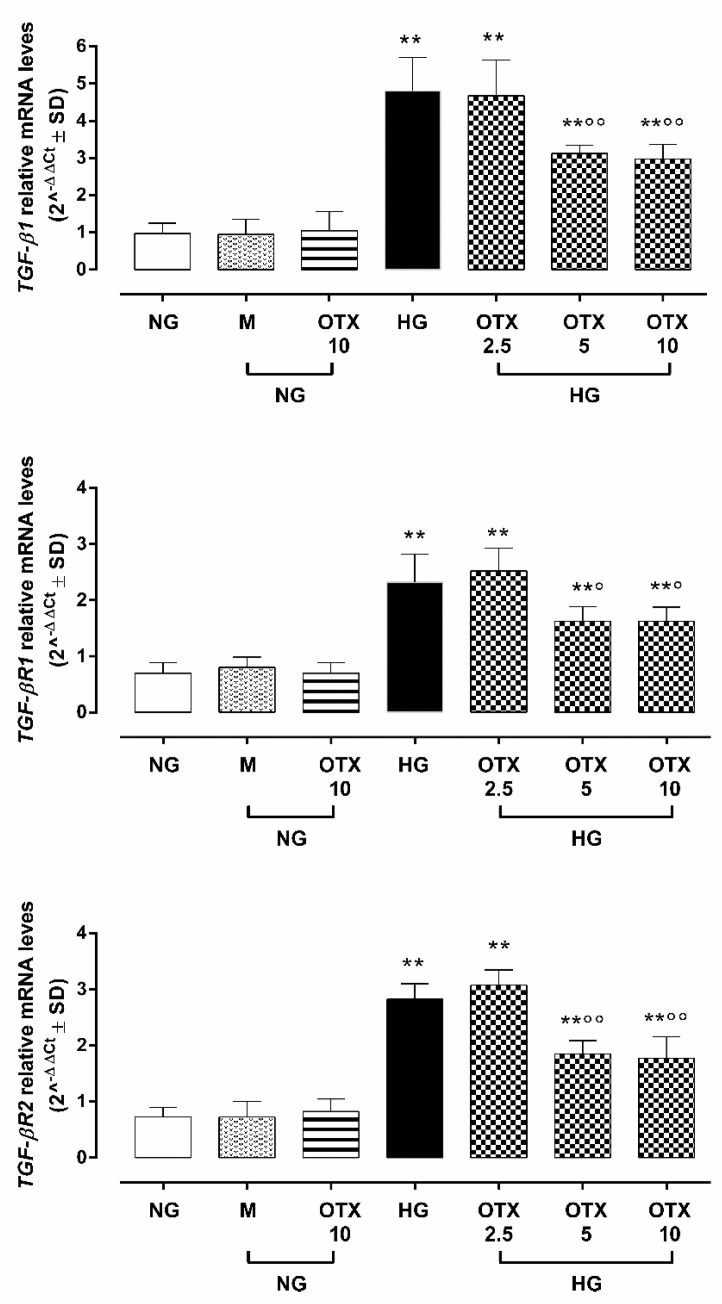
TGF-β signaling in ARPE-19 cells. Cell *TGFβ1*, *TGFβR1*, and *TGFβR2* relative mRNA levels, reported as 2^^−ΔΔCt^ ± SD and quantized by using *GAPDH* as control. N = 9 observations per group (three independent experiments, each performed in triplicate). ** *p* < 0.01 vs. NG; ° *p* < 0.05 and °° *p* < 0.01 vs. HG; NG, normal glucose (5 mM) + vehicle; M, mannitol (30 mM); HG, high glucose (35 mM) + vehicle; OTX 2.5, OTX 2.5 µM; OTX 5, OTX 5 µM; OTX 10, OTX 10 µM.

**Figure 5 molecules-27-04785-f005:**
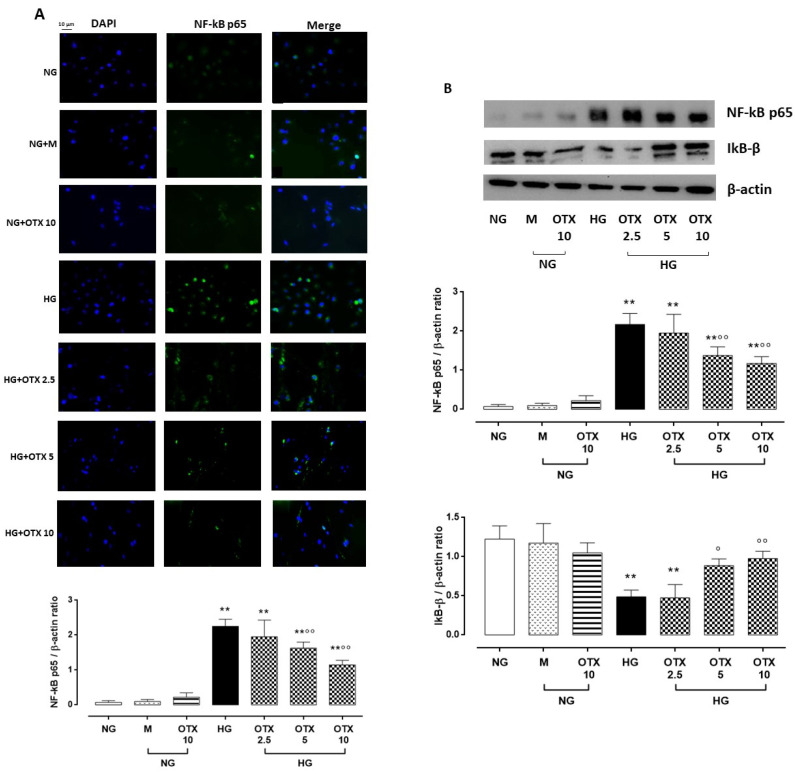
NF-kB and IkB-β levels in ARPE-19 cells. (**A**) Representative immunofluorescence images of NF-kB-positive staining, reported as percentage of NF-kB-positive cells/total cells counted ± SD. Magnification = 10×; scale bar = 10 µM. (**B**) Representative Western blotting dots of NF-kB p65, IkB-β, and β-actin. Protein levels, quantified as densitometric units ± SD, are reported as NF-kB p65/β-actin and IkB-β/β-actin ratio. N = 9 observations per group (three independent experiments, each performed in triplicate). ** *p* < 0.01 vs. NG; ° *p* < 0.05 and °° *p* < 0.01 vs. HG; NG, normal glucose (5 mM) + vehicle; M, mannitol (30 mM); HG, high glucose (35 mM) + vehicle; OTX 2.5, OTX 2.5 µM; OTX 5, OTX 5 µM; OTX 10, OTX 10 µM.

**Figure 6 molecules-27-04785-f006:**
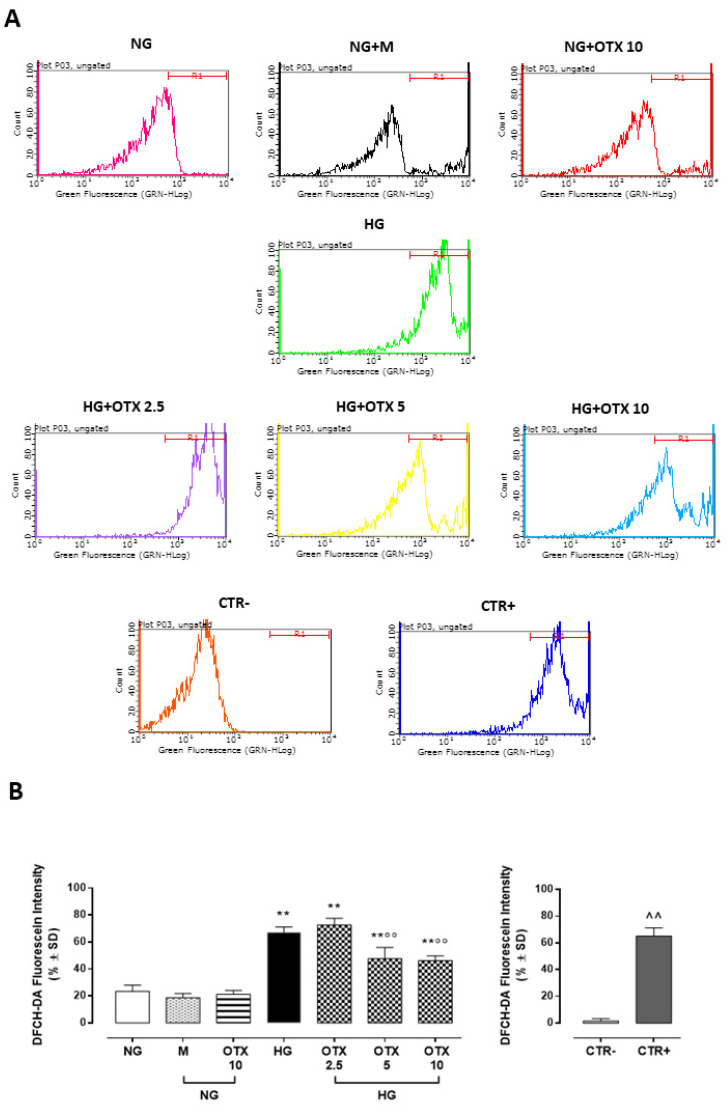
ROS production in ARPE-19 cells. (**A**) Total intracellular ROS levels assayed with 2′,7′-dichlorodihydrofluorescein diacetate (DCFH-DA) probe and measured with a fluorometric plate reader (excitation = 485 nm; emission = 530 nm). (**B**) DCFH-DA levels, reported as percentage of fluorescein intensity ± SD of 9 observations per group (three independent experiments, each performed in triplicate). R1 region = DCFH-DA-positive cells. ** *p* < 0.01 vs. NG; °° *p* < 0.01 vs. HG; ^^ *p* < 0.01 vs. CTR–; NG, normal glucose (5 mM) + vehicle; M, mannitol (30 mM); HG, high glucose (35 mM) + vehicle OTX 2.5, OTX 2.5 µM; OTX 5, OTX 5 µM; OTX 10, OTX 10 µM; CTR–, negative control (5% FBS without DCFH-DA); CTR+, positive control (H_2_O_2_ 100 µM) used to gate R1 region.

**Figure 7 molecules-27-04785-f007:**
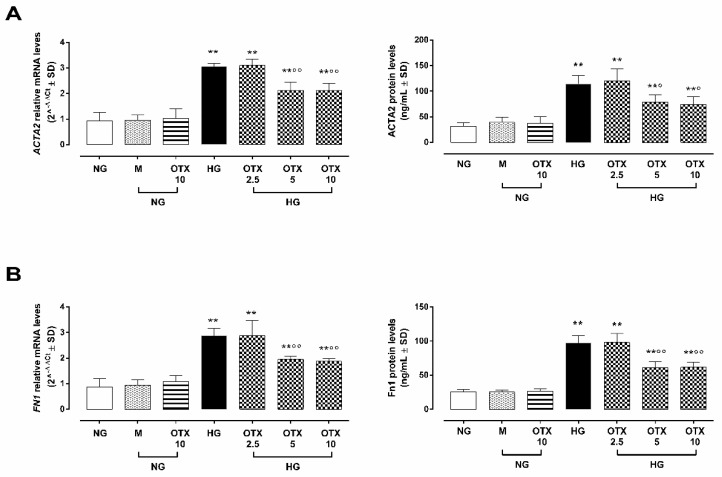
EMT markers in ARPE-19 cells. (**A**) ACTA2 and (**B**) Fn1 mRNA and protein levels. *ACTA2* and *Fn1* relative mRNA levels are reported as 2^^−ΔΔCt^ ± SD and quantized by using *GAPDH* as control. ACTA2 and Fn1 protein levels are reported as ng/mL ± SD of 9 observations per group (three independent experiments, each done in triplicate). ** *p* < 0.01 vs. NG; ° *p* < 0.05 and °° *p* < 0.01 vs. HG; NG, normal glucose (5 mM) + vehicle; M, mannitol (30 mM); HG, high glucose (35 mM) + vehicle; OTX 2.5, OTX 2.5 µM; OTX 5, OTX 5 µM; OTX 10, OTX 10 µM.

## Data Availability

The data presented in this study are available in the present article and its Appendix A.

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
