# Peer review of "Effects of the Calix[4]arene Derivative Compound OTX008 on High Glucose-Stimulated ARPE-19 Cells: Focus on Galectin-1/TGF-β/EMT Pathway"

_molecules, 2022, doi:10.3390/molecules27154785_

Round 1

Reviewer 1 Report

The manuscript molecules-1813471 "Effects of the Calix[4]arene Derivative Compound OTX008 on High Glucose-Stimulated ARPE-19 Cells: Focus on Galectin-1/TGF-β/EMT Pathway" by Hermenean and co-workers et al. describes the study of the effect of well-know macrocyclic ammonium compound, i.e. calix[4]arene OTX008, on high glucose-stimulated ARPE-19 cells. The authors have interesting experimental results, so I think that this paper will be of interest to the readers of Molecules.

Questions and comments:

1) I recommend the authors to strengthen the Introduction part about biological applications of calixarene derivatives. Recent review articles on this topic should be added. For example, Molecules 2020, 25(21), 5145; Supramol. Chem. 2020, 32(3), P. 178-206; Angew. Chem. Int. Ed. 2020, 60(6), 2768-2794.

2) I recommend adding the structural formula of calixarene OTX008 (or PTX-008) to the manuscript or supplementary materials.

3) What does the phrase "Considering that OTX is a synthetic of 3 compounds…" (line 275) mean?

4) The "Conclusions" part should be added. In this part, it is also necessary to add information on the possible practical use of the obtained results.

5) The manuscript contains many abbreviations. A list of abbreviations must be added.

6) I recommend comparing the results obtained by the authors with previous results obtained by other scientific groups.

7) Minor changes:

- lines 246, 253. Maybe "OTX0088" should be "OTX008"?

- in the text there are 2 abbreviations of galectin-1 (Gal-1 and gal-1).

- no reference to the Figure S1in the text of the manuscript.

Author Response

Reviewer 1

The authors would like to thank this Reviewer for appreciating the novelty of the paper. The answers to the criticism of this Reviewer are below. Please, consider that all changes are highlighted in the Tracked Changes version of the Manuscript.

1) Thanks for the suggestion. The Introduction has been strengthen by adding the review articles indicated by this Reviewer, with a brief description of both topics.

2) According to this Reviewer’s suggestion, OTX008 structure has been added to the Supplementary Materials as Figure S1. Moreover, OTX008 molecular formula has been added to 4.1 section.

3) The phrase "Considering that OTX is a synthetic of 3 compounds…" (line 275) has been corrected. Thanks

4) Thanks for the suggestion. A conclusion has been added.

5) Thanks for the suggestion. A list of the abbreviations has been added before the Introduction section. added.

6) Thanks for the suggestion, the results obtained have been compared with the previous work by Abu El-Asrar et al, 2020 (ref 17), testing OTX008 on HMRECs.

7) Minor changes:

- lines 246, 253.  "OTX0088" has been corrected "OTX008;

- gal-1 abbreviation has been corrected in Gal-1;

- reference to the Figure S2 (previous Figure S1) has been added to the text of the manuscript.

Reviewer 2 Report

In this study, Authors have analyzed the cellular expression of Gal-1 in physiological or high glucose conditions on ARPE-19 cell line highly involved in the diabetic retinopathy (DR)-induced fibrotic process and have analyzed the effect of the selective galectin-1 inhibitor OTX008 on DR.

By proper and effective methods Authors show that OTX008 improved cell viability and markedly reduced galectin-1 protein expression in cells exposed to high glucose, down-regulated TGF-β/, NF-kB p65 levels and ROS content.

Therefore, they conclude that the inhibition of galectin-1 by OTX008 in DR may preserve retinal pigment epithelial cell integrity and functionality, by reducing their pro-fibrotic phenotype and epithelial-mesenchymal transition phenomenon induced by diabetes.

 The work seems well conducted and the results support the rationale.

I have only minor concerns:

 1.       Given the name of the journal, I would show the chemical structure of the active molecule under study, OTX008.

 2.       Since Authors state that OTX008 binds Gal-1 and leads to its oxidation and proteosomal degradation, probably explaining the results of Gal-1 reduction reported in Fig.3, I wonder if they have run the test or counter test using the proteasomal inhibitor MG132. They should not observe any decrease or less than that shown in the figure.

Author Response

Reviewer 2

Many thanks to this reviewer for appreciating the paper. The answers to the criticism of this Reviewer are below. Please, consider that all changes are highlighted in the Tracked Changes version of the Manuscript.

  1. According to this Reviewer’s suggestion, OTX008 structure has been added to the Supplementary Materials as Figure S1. Moreover, OTX008 molecular formula has been added to 4.1 section.
  2. The question raised by the Reviewer on the use of MG132 has been carefully taken in consideration by the authors for future deeper investigations. However they point out the hypothesis reported in Discussion about the proteosomal degradation of Gal-1 has been demonstrated by Al-Obaidi, N. et al. FASEB J 2019, 33, 373-387. Therefore, in order to avoid any missinterpretation the authors rephrased the sentence starting with “OTX0088, being a well-established inhibitor of Gal-1…”.

Round 2

Reviewer 1 Report

I thank the authors for answering my questions and improving the manuscript.

Reviewer 2 Report

The manuscript has been sufficiently improved to warrant publication in Molecules since the authors responded comprehensively to the issues raised by the reviewer.